# Animal Models of Depression: What Can They Teach Us about the Human Disease?

**DOI:** 10.3390/diagnostics11010123

**Published:** 2021-01-14

**Authors:** Maria Becker, Albert Pinhasov, Asher Ornoy

**Affiliations:** 1Adelson School of Medicine, Ariel University, Ariel 40700, Israel; mariabe@ariel.ac.il; 2Department of Molecular Biology and Adelson School of Medicine, Ariel University, Ariel 40700, Israel; albertpi@ariel.ac.il; 3Hebrew University Hadassah Medical School, Jerusalem 9112102, Israel

**Keywords:** depression, anxiety, animal models, behavioral tests, chronic and acute stress, social behavior

## Abstract

Depression is apparently the most common psychiatric disease among the mood disorders affecting about 10% of the adult population. The etiology and pathogenesis of depression are still poorly understood. Hence, as for most human diseases, animal models can help us understand the pathogenesis of depression and, more importantly, may facilitate the search for therapy. In this review we first describe the more common tests used for the evaluation of depressive-like symptoms in rodents. Then we describe different models of depression and discuss their strengths and weaknesses. These models can be divided into several categories: genetic models, models induced by mental acute and chronic stressful situations caused by environmental manipulations (i.e., learned helplessness in rats/mice), models induced by changes in brain neuro-transmitters or by specific brain injuries and models induced by pharmacological tools. In spite of the fact that none of the models completely resembles human depression, most animal models are relevant since they mimic many of the features observed in the human situation and may serve as a powerful tool for the study of the etiology, pathogenesis and treatment of depression, especially since only few patients respond to acute treatment. Relevance increases by the fact that human depression also has different facets and many possible etiologies and therapies.

## 1. Introduction

Depression is probably the most common behavior-debilitating disease and is the most prevalent psychiatric disease among the mood disorders [1,2,3,4]. This widespread and chronic psychiatric illness can affect thought, mood and physical health causing serious disabilities. It is generally characterized by low mood, sadness, insomnia, lack of energy and no joy in life [5]. It is also a leading cause for suicidal attempts [6,7,8].

Depression affects up to 10% of the population and is about twice as common in women compared to men [3,9,10]. It may affect adolescents and adults and is more prevalent among elderly people [3,11]. Several medical conditions such as diabetes, chronic pain, cancer and emotional stress may increase the prevalence of depression [3,12] Depression is more common during pregnancy and post-delivery, affecting about 8% in the first trimester and 12–13% in the second and third trimester and in the peripartum period [13,14].

The etiology of depression is still poorly understood. Genetic, epigenetic and environmental factors are apparently involved, with distinct changes in the hypothalamic–hypophyseal–adrenal axis [3,15]. Depression is characterized by monoamine synaptic imbalance [16,17]. Indeed, pharmacological treatment uses various drugs that raise the synaptic concentrations of serotonin or norepinephrine [18,19]. In spite of the fact that depression has important genetic elements, no specific genomic changes or polymorphisms can be directly linked to its etio-pathology [3,20,21]. Environmental factors such as stress, traumatic events and chronic pain seem to be important contributory factors [22]. It is therefore believed that depression results from the interaction of genetic and environmental factors [3,16]. Treatment is mainly based on modulation of brain monoamines by appropriate drugs and psychological support, especially Cognitive Behavioral Therapy (CBT). However, often treatment is ineffective, with aggravation of symptoms [23].

Depression and anxiety are highly comorbid [24], either occurring concomitantly or sequentially. Pharmacological treatment is often similar, as selective serotonin reuptake inhibitors (SSRIs) are effective for both problems, but sometimes different drugs or drug combinations must be used with emphasis on the disorder with the most severe symptoms. Depression and anxiety are also closely associated by sharing similar etiologies and genetic predisposition [25].

Like many other diseases, the understanding of the etiology and pathogenesis of depression can benefit from animal models. Moreover, appropriate animal models can also help in the search for effective treatment. Hence, genetic, epigenetic and environmentally induced animal models that reproduce symptoms typical to human depression have been developed, especially in rodents, to help the investigators better understand human depression. The purpose of the present review is to briefly discuss the more common animal models of depression and stress their strength and weakness.

## 2. What Are We Looking for in Animal Models of Depression?

Animal models may serve several purposes: first, they may enable a better understanding of scientific processes [26]. However, when evaluating animal data, it is important to remember that animals are not human beings, and many physiological and biochemical processes are different. Hence, when we evaluate the animal data we need to know how close is the etiology and pathogenesis in the animal model to that of the human disease and whether the measures of symptoms are reliable enough to resemble those of the human situation. This is especially important when we use neurobehavioral tools for measurement. Indeed, the modeling of depression in animals was initially based on the evaluation of abnormal social behavior, motivation, working memory, emotion, and executive functions.

Generally, animal models must meet three basic values: face validity, when animals recapitulate disease phenotype in a similar way to depressed humans; etiologic (construct) validity (relevance), when pathophysiological processes in animals are similar to those that cause a disease in humans; and predictive validity (pharmacologic sensitivity), when animals respond to medicines that are effective to treat depressed humans [27,28]. Unfortunately, many of the animal models of depression do not fulfill all these principles. Commonly, the construct validity in contemporary established animal models of depression is different from the real state of human depression because of the multiple intrinsic and extrinsic factors that may trigger the onset of depression and the fact that the exact pathophysiological mechanism still has to be elucidated. The requirements for animal models were revised, and additional validity criteria were introduced: homological validity; pathogenic validity; mechanistic validity; convergent validity; discriminant validity; internal and external validity [29,30].

Well-established animal models of depression were developed by applying different methodologic approaches, such as stressful factors or genetic manipulations, which triggered the appearance of depressive-like phenotype in rodents. Some animal models of depression were developed for testing the efficacy of newly developed antidepressant drugs or compounds. Some models, especially transgenic mice, serve to explore genetic and molecular pathogenic mechanisms. Similar to the situation in other animal models of human diseases, there seems to be no single animal model that can be used for all purposes of research on depression.

## 3. Testing of Depression-Like Behavior in Animals

The testing of behavior in rodents is usually based on the observation of certain behavior traits that can be considered analogous to the symptoms of depression in humans [31,32,33,34,35,36]. Therefore, when describing animal models of depression, we use the description “depressive-like behavior”. Symptoms of major depressive disorder (MDD) such as anhedonia and depressed mood, apathy, sleep disturbances, weight/appetite changes, psychomotor changes, and other comorbid conditions, such as anxiety and social isolation, may be easily evaluated in animals [37,38]. Other symptoms of depression in humans, such as feelings of sadness, guilt, or suicidal thoughts, cannot be simulated in rodents or other animal models.

### 3.1. Evaluation of Despair-Like Behavior

Feelings of despair or learned helplessness are modeled in rodents by placing an animal in an unpleasant, unescapable situation and evaluated by measuring certain behavioral traits by which the animal tries to escape that environment. The gold standards for despair testing are the Forced Swim Test (FST) [31], when the animal is placed in a water tank, and the Tail Suspension Test (TST) where the animal is suspended by its tail [32]. Initially, animals actively swim or struggle, and further, the animals begin to show periods of immobility, which then gradually increase. The increase in the immobility time is considered as depression-like behavior.

### 3.2. Evaluation of Apathy-Like Behavior

Apathy is considered a deficit in voluntary, goal-directed behavior [39]. In rodents, signs of apathy are manifested by impaired nest formation, unkempt hair due to impaired self-care, scanty maternal care, decreased social interest and decreased interest in novelty. These are considered as a depressive phenotype [37,40]. One of the ways to evaluate motivational and self-care behavior in rodents is a Splash-test [35,41]. The frequency and the duration of grooming are recorded during 5 min after sprinkling of 10% sucrose solution on the animals back coat, usually in its home cage. The viscosity of sucrose solution imitates dirt on the rodent fur and triggers grooming by licking or scratching. The delay in time between sprinkling and initiation of grooming and decreased frequency of grooming are considered as reduced motivational and self-care behaviors associated with depression-like phenotype.

### 3.3. Evaluation of Anxiety-Like Behavior

Anxiety commonly co-occurs with MDD and is evaluated in animals by studying the approach–avoidance conflict, the contradiction between their innate curiosity to explore a novel environment and their preference to be in closed/dark instead of open/illuminated places [42]. The commonly used test for anxiety evaluation is Elevated Plus Maze (EPM) or Zero Maze [33,43,44], where longer latencies to enter and/or shorter time spent in the open arms are interpreted as anxiety-like behavior. The open field test is another often used test, to assess both anxiety and locomotor activity [45]. The animal is placed in an open box, where its anxiety level is defined by the latency to enter and time spent in the center of the arena and the total traveled distance. These are taken as measures of locomotor activity. Anxiety may also be tested by novelty suppression of feeding (NSF) [46], conflict between the drive to eat food pellets in open/light space and the preference for closed/dark space [47]. The tree chamber test [48] for testing social interaction in rodents can also be used to measure level of anxiety.

### 3.4. Evaluation of Anhedonia-Like Behavior

Anhedonia means the loss of the ability to experience pleasure in things that previously seemed pleasant to a person. In rodents, anhedonia is usually expressed as a decreased preference for consumption of sweets (sucrose) [34], intracranial self-stimulation (ICSS) [49], preference for novel objects or situations, or frequency of sexual interactions [38].

### 3.5. Evaluation of Impaired Cognitive Function

Memory deficits and learning difficulties are often observed in depressive patients [50]. In rodents, cognitive behavior may be evaluated by the standard Morris Water Maze [51]. The ability of the rodent to learn to find a hidden platform beneath water reflects the rodent’s hippocampal-dependent spatial memory and learning. Mice taking a longer time and a more complex trajectory of swimming path to find the platform are considered cognitively impaired with memory deficit. The Y-maze test is a widely used test for working and short-term memory. Two paradigms, spontaneous alternation and short-term memory, may be studied using the Y-maze, depending on the test-design protocol [52,53]. The percentage of spontaneous alternations is based on the frequency of complete alternations between the three arms.

### 3.6. Evaluation of Social Interaction/Social Withdrawal

Social withdrawal is one of the many symptoms in depressed patients [54] and may be easily measured in rodents by using the three-chamber paradigm test known as Crawley’s sociability and preference for social novelty test [36,55]. This test is based on the spontaneous exploration by the mouse of any of the three chambers of the box, including indirect contact with one or two stranger mice. Sociability is defined as the preference to spend more time with a stranger, a different mouse, compared to the time spent in an empty chamber. This is recorded in a second session after the first habituation session. Mice spending less time near the stranger mouse are considered as having depression-like symptoms. However, social impairment is more obvious in mice with autism-like behavior, but can also be used to study depression-like behavior [56,57].

## 4. Models of Depression

As stated above, there are many different models of depression. The most widely used are environmentally induced and genetically based models. We will therefore discuss these different models at some length. Table 1 summarizes the different rodent models for depression, their strengths and weaknesses.

### 4.1. Environmentally Induced Models of Depression

The etiology of depression is primarily associated with acute and chronic stressful factors [200]. In some cases, acute or chronic stress leads to depression and other psychiatric disorders, but not all individuals become depressed; some individuals are resilient to stress [201,202]. The precise mechanisms of resilience or susceptibility to stressors is far from being elucidated [202]. It seems, therefore, that genetic predisposition, adaptation pathways in brain circuits mediated by various neurotransmitters [203,204,205] and hormones [206], and the further response of individuals to stressful stimuli underlie successful coping with stress. It has been proposed that a dysfunction in the stress hormone, i.e., corticosterone regulation, is causally linked to depression [207,208]. Chronic and persistent stress impairs the hypothalamic–pituitary–adrenal axis, alters the immune system and induces various pathophysiological events detected in patients with depression [38].

Some established animal models will be described regarding applied stress factors (stressors). Early-life stress (prenatal stress, maternal separation, post-weaning social isolation), chronic-mild stress or social defeat stress models are the most validated and widely used models of depression, inducing a depression-like phenotype in experimental animals (Table 1).

#### 4.1.1. Early-Life Stress

Early-life stress, when experienced during critical periods of development, such as prenatal, early postnatal and adolescence, may produce permanent changes to neural circuits with long-term negative sequelae [209,210]. Individual development (ontogenesis) of different parts of the brain occurs at different times during these periods, thus broadening the temporal window of vulnerability and the number of developmental processes [210,211,212,213].

##### Prenatal Stress Modeling

Maternal stress during pregnancy may affect offspring neurodevelopment and may lead to later appearance, at adulthood, of a psychiatric disorder, such as anxiety and depression [214,215,216]. Stress hormones, corticotropin-releasing hormone (CRH), catecholamines and glucocorticoids are transported via the placenta, reach, and affect the fetal brain [217]. Usually, gestational stress is induced daily by various stressors: combination of restraint, 24-h light disturbance, sleep deprivation, forced swimming or crowding in pregnant dams throughout gestation [58,218,219,220]. The prenatal stress paradigm causes the depression-like phenotype in rodents, demonstrating good face validity value. Prenatally stressed offspring show prolonged immobility in the FST test [59,60,61] and in the tail suspension test [58] as well as reduced preference to sucrose solution [58]. Prenatal stress also induces anxiety-like behavior in adolescent offspring measured by decreases in the time spent and number of entries in the open arm in the EPM test [58,61,62]. These changes were more pronounced in females than in males [59,221]. Prenatal stress paradigm has good construct validity: increased plasma corticosterone levels [60,63], altered brain-derived neurotrophic factor (BDNF) transcripts and protein expression [58,61] and morphological changes in dendritic complexity in the offspring hippocampus [218,222,223].

One important consideration for using this prenatal stress paradigm are differences in the rate and time of brain ontogeny between rodents and humans. These differences should be taken into consideration when translating the obtained results to the human situation [64,65,66] (Table 2).

##### Maternal Separation and Models of Early Life Stress

Many studies on various species such as humans, nonhuman primates and rodents indicate that early adverse postpartum care (or stress at an early-life stage) increases the risk of depression in adulthood [224,225,226,227,228]. Maternal separation is one of the most studied animal models regarding the influence of early-life stress events on the development of depression later in adulthood [72]. The brain developmental status of rats and mice at the time of birth corresponds to that of the end of the second and early third trimester of the human fetus and is characterized by elevated synaptic plasticity, neurogenesis in the cerebellum and hippocampus and continued increase in the size of the cerebral hemispheres [64,66]. Bayer et al. [67] compared the developmental stages of the human fetal brain to that of the rat embryo and fetus and found, for example, that the development of the rat brain on day 21 before parturition is comparable to the human fetal brain in mid-pregnancy. Hence the early postnatal brain development in rats resembles that of the third trimester human fetus. Generally, the major phases of brain development in rodents and human are similar, but the timeline is drastically different. Similar brain developmental events last for days in rats and mice, and for weeks in humans [66].

Table 2 shows the prenatal and postnatal time differences in brain development between mice/rats and humans. The developmental stages in rats differ from those of mice in the range of 1–2 days.

The early postnatal period reflects a critical window of brain development, during which the brain is particularly vulnerable to harmful events such as stress [66]. Hence, mouse offspring during the early postnatal stage are highly dependent on maternal care. Early separation from the mother is highly stressful, influencing the development of the biological and behavioral phenotype of the offspring at adulthood. Usually, rodent pups are separated from their mothers for 3 h daily from the 2nd to the 12th postnatal day, modeling maternal neglect. Another approach to induce neonatal stress is limited nesting and bedding material in the cages, which models inappropriate maternal care. Later, upon reaching maturity, the animals’ behavioral phenotype is evaluated. Generally, early maternal separation causes learning and memory deficits, depressive and anxiety-like behavior [67] in open field and elevated plus maze tests in either male or female mice [68], reflecting face validity. These behaviors were alleviated by fluoxetine pretreatment at adolescence, but not by chronic fluoxetine treatment at adulthood [74] in correspondence to predictive validity. Neonatal stress causes a decrease in BDNF, an increase in plasma corticosterone levels and a decrease in corticotrophin releasing factor (CRF) signaling pathways [69,70,71] demonstrating construct validity value.

However, this early stress effect inducing a depression-like phenotype is inconsistent [30,40,72] and depends on the mouse strain [73]. In some studies, the opposite or no effect were also observed [73,229,230]. Tractenberg et al., in their systematic review of 94 mouse studies [72] commented that different methodologies of maternal separation were implemented in various studies, leading to variations in the degree of stress exposure for the pups and, later, to uncertain depressive-like behavior. The weakness of this model is the inconsistency of the long-term outcome following maternal separation and the dependence on the specific mouse strain. In addition, the relevance of using infant pups whose brain developmental status corresponds to a prenatal human stage is in doubt.

These depression-like animal models of prenatal and postnatal stress are highly useful in the research and elucidation of the epigenetic mechanisms underlying the consequences of gestational stress or early-life stress on later depression and anxiety. The understanding of the link between early-life aversion and the development of stress resilience or stress susceptibility may provide new insights into biological targets and in the development of new therapeutics for treatment or prevention of depression.

##### Post-Weaning Social Isolation Stress

Adolescence is initiated by central activation of the hypothalamic–pituitary–gonadal (HPG) axis resulting in stimulation of steroid hormone synthesis. These maturation processes, particularly judgmental or behavioral control, are accompanied by enormous sensitivity to social stress and rewarding stimuli [231]. Kessler et. al. [213,232] reported the high prevalence in the onset of mental disorders such as anxiety, depression, eating disorders and schizophrenia during adolescence.

Post-weaning social isolation (isolation rearing) in rodents is one of the most commonly investigated models of social isolation stressors as a risk factor for anxiety, depression and substance use disorders (SUDs) in adolescence, and later in adulthood (reviewed in [83,84,85,86,233]). Raising rodents in conditions of constant social isolation (SI) from weaning, depriving them of the opportunity for social play and disrupting the establishment of stable social hierarchies leads to behavioral changes influencing various psychiatric conditions such as depression, bipolar disorder and schizophrenia [27,76]. Usually, animals on post-natal days (PD) 21–28 are housed alone in a home-cage, maintained in social isolation for 3–8 weeks [234] and are not handled more than once a week (to change bedding material). However, isolated animals may have some visual, auditory, and olfactory contact with other isolated animals and those of their group-housed counterparts maintained in the same unit. Rats growing in post-weaning social isolation (SI-reared) develop an “isolation-induced stress syndrome” with excessive reactivity to handling, anxiety-like behavior and high emotionality [75,235]. Male and female SI-reared rats demonstrate good face validity of the depressive-like phenotype with increased motor locomotion [75,76,77] and decreased time spent in the arena center [78] in the open field test, and increased anxiety-like behaviors tested by the elevated plus maze [79,80,81]. These effects are irreversible by resocialization [236], as observed in males but not in females [237,238]. SI reared rodents demonstrated despair-like behavior expressed as increased immobility time in the FST [76,239] that was reversed by chronic fluoxetine treatment [82], indicating a good predictive validity. SI-reared rats exhibited apathy-like behavior expressed as diminished grooming time and frequency in the splash test [75], mainly in males. In addition, some studies showed that SI-reared rats, either male or female, demonstrated increased adult social interaction [77,82,240].

SI-reared induced abnormal aggressive behaviors in male rats are tested by the resident-intruder test. This abnormal aggressiveness was named “behavioral fragmentation pattern” because of rapid fluctuations from one behavior to another, from abnormal attack to social interaction and to increased defensive upright, flight and freezing behaviors [241]. This abnormal aggressive behavior was attenuated by resocialization with concomitant chronic fluoxetine treatment [241]. Post-weaning SI paradigm also demonstrates good construct validity expressed as lower corticosteroid plasma levels [242,243], enhances synaptic plasticity of NMDAR-mediated glutamatergic transmission in the Ventral Tegmental Area (VTA) [244] and altered serotonergic and adrenergic systems [83,245] along a neuroaxis.

The weakness of this mode is that post-weaning SI, inducing long-lasting or even permanent depressive-like and anxiety-like behaviors, is rather considered as a model of reward-associated behaviors such as drug preference and alcohol consumption in adulthood schizophrenia and bipolar disorder [76,81,85,87,88].

When the social isolation stress paradigm is applied during adulthood it triggers in rodents both anxiety- and depression-like behaviors [89,90,91,92,93,94], evaluated by open field, EPM, FST, TST and anhedonia, expressed as reduced sucrose drinking and alterations in sexual reward behavior. Chronic fluoxetine treatment attenuated depressive-like behavior in the FST and TST in SI female mice but had no effect on anxiety-like behavior [93]. The altered behavior was accompanied by reduced plasma corticosteroids levels measured in SI adult mice [92,93], and lower expression levels of the BDNF and neuroplasticity-related genes in both hippocampus and prefrontal cortex [90]. It seems that the social isolation model of depression has good face, predictive and construct validity. However, the study reports are inconsistent and demonstrate strain-dependent and test-specific effects of depressive and anxiety-like behaviors [95,96,97].

#### 4.1.2. Adulthood Stress

##### Learned Helplessness in Rats/Mice (Acute Stress)

The Learned Helplessness (LH) paradigm is widely used to explore the effect of unpredictable and uncontrollable stress applied during adulthood in rodents, who further develop an inability to cope with aversive but escapable situations. At the beginning this paradigm was used to induce depression-like phenotype in dogs [246] and was later adopted to rats [247,248,249] and mice [250]. LH paradigm is considered a model which manifests good similarity to the symptoms of depression, with construct and predictive validity, and cognitive and neuroendocrine impairments. In the learned helplessness paradigm, animals first receive several electric shocks on their feet in a closed chamber. Then, the subjects are placed in another chamber with a grid floor and receive a mild shock with the possibility of escape. Rodents that have not been previously exposed to the unescapable shock are commonly able to escape quickly from the shock, whereas animals previously exposed to the learned helplessness paradigm frequently fail to avoid shock [248,251]. This is considered as helplessness and indicates a depressive-like phenotype. Learned helplessness is reversed by antidepressant treatment [101,102,103]. These animals also demonstrate reduced weight, changes in locomotor activity (decreased locomotion or hyperactivity), sleep disturbances, decreased motivated behavior and anhedonia, but normal learning and memory [98,99]. Learned helplessness is associated with elevated glucocorticoid levels and reduced negative feedback responses of the hypothalamus–pituitary–adrenal (HPA) axis [100]. The learned helplessness paradigm causes an increase in pro-inflammatory cytokine levels, such as IL-1β, IL-6, TNF-α, INF-γ and G-CSF in depressive-like animals [252,253].

This method of inducing depression-like behavior is not effective in all strains of rodents. The selective breeding of helpless lines from Harlan Sprague–Dawley outbred rats at State University of New York (SUNY) at Stony Brook SUNY, was developed as an effort to achieve a higher yield of helpless animals for learned helplessness following inescapable shock-training. Two variants were established: the congenitally learned helpless (cLH) rats exhibiting a helpless phenotype without exposure to uncontrollable shock, and a congenitally, not learned, helpless (cNLH) strain resistant to the effects of inescapable shock [100]. The weakness of this model is the inconsistent response to the above-mentioned stress and weak etiologic validity, because the exact role of the genetic construct of the responding animal is as yet unknown.

##### Chronic Mild Stress (CMS)

Willner et al. developed a paradigm of chronic mild stress (CMS), where different types of mild stressors are alternated [104,105]. CMS induced a depression-like phenotype with good face, construct and predictive validity. Usually, to induce CMS the animals are exposed for three weeks to a variety of long-term inescapable stressors, such as walking on ice, tube restraining, tail suspension, day and night reversal, tail clipping, and water or food deprivation which are performed every day randomly. Then, behavioral traits assessments are carried out. The most obvious feature of CMS that the animals manifest is anhedonia [106,107], demonstrated by sucrose intake and preference, which is reversed by chronic, but not by acute, antidepressant treatment [110]. Rodents exposed to CMS also exhibited hyperlocomotion in the open field test [107], shorter time spent in the open arm in EPM [108], and prolonged immobility time in the FST [109]. The CMS paradigm induces altered regulation of the hypothalamus–pituitary–adrenal (HPA) axis, changes in serotonergic, noradrenergic and dopaminergic systems, and reduced hippocampal BDNF level (details are reviewed by Hill et al.) [111]. However, the data on the depressive-like phenotype are inconsistent and the reproducibility of this model among different research centers varied [105]. Different studies used different methodologies and applied stressors, resulting in variations in the degree of stress on the mice and in undefined depressive behavior.

##### Repeated Restraint Stress (CRS)

The CRS model also represents the depression model with good validity and is used widely in preclinical research to explore the effect of chronic psycho-emotional stress [116,117,118]. For chronic restraint stress, usually the animal is placed into a narrow cylindrical restrainer with a nose-hole for ventilation, in which it is unable to move, for 2–8 h daily for 21 consecutive days, followed by behavioral assessment. Rodents exposed to CRS exhibited the depressive-like phenotype in social interaction, anhedonia, increased anxiety and impaired spatial learning [112,113,114,115]. These behavioral features were attenuated by chronic treatment with fluoxetine. In addition, CRS triggered the elevation of serum corticosterone levels [115].

The weakness of CRS models lies in the fact that the induced short-term depression symptoms tend to subside.

##### Chronic Social Defeat Stress

Repeated social stress is the most common etiologic factor that triggers the development of depression in humans [119,120]. The social defeat stress animal model is based on the resident-intruder paradigm and simulates the pathogenesis of depression at the social level. Usually, for induction of mouse models of social defeat, ICR mice are used as the resident and C57Bl/6J (B6) mice are used as the intruder which undergoes the physical and psychological stress [254]. Proven aggressive ICR mice are kept for several days in one compartment in the home cage, separated by a transparent acrylic wall, with small holes that allow the circulation of odors, pheromones and vocalizations. A B6 mouse is placed with an ICR mouse, and typically the ICR mouse attacks the B6 mouse (physical stress). After the attack, B6 is moved to a compartment adjacent to the ICR mouse and stays until the end of the day (psychological stress). On the next day, the B6 mouse is placed with another ICR mouse. Commonly, B6 mice are subjected to 10 daily sessions of physical and psychological stress. Interaction periods are recorded and later analyzed for aggressive, submissive, and exploratory behaviors by social interaction tests. This paradigm consistently yielded a subordinate/submissive phenotype of the intruder mouse in about two-thirds of involved animals. Stress-susceptible mice, experiencing repeated aggression, develop a long-lasting aversion to social contact, a deficit in sucrose preference and increased times of immobility in the FST and TST [121,122]. This may be reversed by enriched housing conditions [254,255,256] and by chronic, but not acute, administration of antidepressants [123,124], demonstrating a good predictive validity. The remaining one-third of animals are resilient to stress and fail to develop social avoidance or anhedonia. The social defeat stress animal model also has good construct validity, where alterations in HPA axis and dopaminergic ventral tegmental area (VTA) function were reported [122,125].

This social defeat model fits well with all the main demands of an animal model, demonstrating excellent etiological, face, and predictive validity [122]. However, this model requires the use of a large number of animals, takes up a lot of space in the animal facility and is also labor intensive. In addition, the social defeat stress imposes ethical concerns including the probability of skin damage to the experimental mouse (intruder), as a result of the attack of the aggressor mouse, which sometimes is a reason for further exclusion of the animal from the study. In addition, this model is pertinent mainly to male mice and to specific strains. As social defeat stress is not well adaptable to female rodents because they are less aggressive, the social instability stress (SIS) model was introduced for female animals [127].

Social instability stress is modulated in female rodents by changing the neighbor members within the cage, and by alternating periods of social isolation and crowding phases [126]. The social instability paradigm distorts the previously established social network and forces the animal to adopt to a new hierarchical rank in each of the crowding phases. Rodents that undergo this paradigm exhibit an anxiety and depressive like phenotype in open field, EPM and social interaction tests, and elevated peripheral corticosterone level [127,128,129] and lower hypothalamic glucocorticoid receptor (GR) expression [129]. The depressive-like behavior induced by the social instability paradigm is reversed by chronic antidepressant treatment with fluoxetine in both males and females [257]. This model mimics the male chronic social defeat stress model in many aspects, except for the gender differences and fewer ethical concerns.

### 4.2. Lesions of the Brain that May Result in Depression

#### Olfactory Bulbectomy (OBX)

Anosmia or loss of odor discrimination is sometimes present in depressed patients [258,259]. This olfactory dysfunction may arise from the reduced volume of the olfactory bulb in depressed individuals [260]. The surgical ablation of olfactory bulbs in rodents is an appropriate depression model with reliable face, construct and predictive validity [130,131,132]. Bilateral olfactory bulbectomy (OBX) in rats is considered a model which represents chronic psychomotor agitated depression. It leads to the “irritable aggression” phenomenon, exhibited by increased attack, struggle, startled and fight responses in social interaction tests [133]. Bulbectomy in rats is followed by loss of odor and pheromone smell and the appearance of various abnormal behaviors such as anhedonia, memory dysfunction, sleep abnormalities and depression-like phenotype [130,134,135,136] which meet face validity. The most typical behavioral trait of OBX rats is hyper-locomotion in a brightly illuminated open field arena, due to paucity of opportunity to adapt to novelty. This hyperactivity is reversed by chronic, but not acute, antidepressant treatment [133]. In addition, OBX rats demonstrated decreased time spent in open arms in the elevated plus-maze [137,138], prolonged immobility time in the FST [138,139] and reduction in sucrose intake in the sucrose preference test (SPT) [140]. OBX rats demonstrated impaired spatial learning and memory deficit in MWM, in the passive avoidance test, and in the 8-arm radial maze. These cognitive impairments are corrected by chronic SSRI treatment [135,141,142]. The abnormal, depressive-like behavioral traits in OBX rats feature altered function of the noradrenergic, serotonergic, cholinergic, GABAergic and glutamatergic neurotransmitter systems (reviewed in [130,143,144]), underlying construct validity.

Accumulation of Beta Amyloid has been observed in OBX rodents, and this model is therefore also applied to explore the molecular mechanism and drug efficiency of Alzheimer’s disease (AD) [261,262].

The weakness of this model is the fact that it is produced by injury to the olfactory bulb cortex, mimics only a limited number of depressed patients and leads to significant cognitive impairment.

### 4.3. Pharmacological Models

#### 4.3.1. Reserpine Induced Depression Model

Reserpine treatment in rodents induces depression-like behavior, that displays construct validity of monoamine dysfunction implicated in the development of depression. Reserpine is a vesicle reuptake inhibitor, which arrests the neurotransmitters (norepinephrine, epinephrine, dopamine and serotonin) reuptake process on the presynaptic membrane and, therefore, facilitates their further degradation by monoamine oxidase. The depletion of catecholamines (norepinephrine, epinephrine, and dopamine) and serotonin (5-HT) leads to morphological changes in the brain of tested animals and has good face validity due to the appearance of the depression-like phenotype [145,146,147,148,157]. This model is widely used in preclinical studies to evaluate the antidepressant effect of new developing drugs and plant compounds [149,150,151,152,153,154] due to the short-term time and good predictive validity [147,155]. However, reserpine treatment produces motor impairments mimicking Parkinson disease dyskinesia, increased nociceptive sensitization and hypothermia and has high mortality rate [148,155,156,157].

#### 4.3.2. Corticosterone Model of Depression

High levels of glucocorticoid administration mimics chronic stress. Corticosterone might be delivered to animals over a period of weeks to months by different methods: administered by subcutaneous injection, by pellet or osmotic pump implantation, or through the drinking water or food [160]. Chronic corticosterone treatment causes various behavioral abnormalities in rodents, such as increased immobility in the forced swim test, decreased grooming, impaired memory in MWM and T maze, anxiety-like behaviors in the open field and light/dark test, and anhedonia expressed by low sucrose preference [40,56,158,159,160]. Chronic treatment with fluoxetine [161,162], and an acute single dose of ketamine [163] reverse this corticosterone-induced depression-like phenotype.

Positron emission tomography with 18F-fluorodeoxyglucose ({18F} FDG) in Long–Evans rats chronically treated with corticosterone revealed decreased metabolic activity in the insular cortex and the striatum, but elevation in the cerebellum and midbrain [263]. Moreover, prolonged corticosteroids treatment led to structural changes in rodents’ brains, such as reduced hippocampal volume and increased volume of amygdala (reviewed in [160]). Chronic corticosterone treatment may induce many metabolic and biochemical changes outside the brain, affecting animal behavior in a different way compared to human depression.

### 4.4. Genetic Models of Depression

#### 4.4.1. Wistar Kyoto (WKY) Model

The Wistar-Kyoto (WKY) rat strain is considered a good animal model of endogenous depression. This rat strain was developed as a normal blood pressure control for the spontaneously hypertensive rat (SHR) by Okamoto and Aoki [171]. Besides normal blood pressure, WKY rats demonstrated hyper-reactivity to stress accompanied with dis-balance of the hypothalamic–pituitary–adrenal (HPA) and hypothalamic–pituitary–thyroid (HPT) axes [169,170,171]. WKY rats developed depressive-like behavior in response to acute stress and chronic mild stress as measured by various behavioral tests, demonstrating good face construct and partial predictive validity. These rats demonstrated an increased immobility in the forced swim test (FST) [164,165], decreased activity in the open field test [166,167] and decreased consumption of a sugar-based solution (enhanced anhedonia) [168]. Depressive-like behavior of WKY rats was slightly attenuated by treatment with tricyclic antidepressants (TCA) [173,174], but was resistant to Fluoxetine, a selective serotonin reuptake inhibitor (SSRI) [175].

WKY rats exhibit altered endogenous brain neurotransmitters, including abnormalities of the monoaminergic, dopaminergic and noradrenergic neurotransmitter systems and thyroid-stimulating hormone systems. The main critical feature of this animal model is HPA dysfunction, manifested by higher plasma content of anterior pituitary adrenocorticotropic hormone (ACTH) [172], decreased levels of serum corticosterone, TNF-α, IL-1β, increased IL-10 [264], and down-regulated glucocorticoid receptors (GR) in the hippocampus. Hence, this model exhibits significant differences in comparison to the human situation as it presents an abnormal HPA and HPT functions.

#### 4.4.2. Genetically Selected Flinders Sensitive Line (FSL) Rat Model

Flinders Sensitive Line (FSL) and Flinders Resistant Line (FRL) rats were bred from Sprague-Dawley (SPD) rats at Flinders University in Australia. They produced strains with increased (FSL) or decreased (FRL) sensitivity to the cholinesterase inhibitor, an organophosphate anticholinesterase agent, diisopropyl fluorophosphate (DFP) [265]. FSL rats have higher levels of muscarinic receptors in the striatum and hippocampus compared to FRL rats [266], which undelay their sensitivity to cholinesterase inhibitors. Similarly, depressed humans are more sensitive to cholinergic agonists [267]. This line of rats exhibits behavioral features characteristic of depression, and respond to chronic, but not acute, antidepressant treatment [177,178], with good predictive validity. The FSL rats have been proposed as a rat model of human depression with psychomotor retardation, because they are hypoactive in open field and exhibit prominent immobility in the forced swim test [176] reflecting good face validity. FSL rats also exhibit reduced appetite and psychomotor dysfunction, sleep and immune abnormalities, but preserve normal hedonic state and cognitive function [177,268]. FSL rats have good construct validity as neurochemical and/or pharmacological studies demonstrated changes in the cholinergic, serotonergic, dopaminergic, NPY, and circadian rhythm systems, with normal functions of noradrenergic, HPA axis and GABAergic systems [177]. FSL exhibit anhedonia only when exposed to chronic mild stress [269], implying the potency of this model to elucidate the impact of environmental factors and gene predisposition in MDD pathogenesis.

This genetic model of endogenous depression represents good validity and serves not only to investigate the pathogenic mechanisms that underlie the morbidity of MDD but is also an appropriate model to evaluate the efficacy and safety of antidepressant agents in pregnant DAMs and their offspring during early life and later adulthood [179]. The main weakness of this model lies in the fact that there are many changes in different brain neurotransmitters which are not necessarily present in people with depression.

#### 4.4.3. Model of Selective Breeding for Depressive-Like (Submissive) and Manic-Like (Dominant) Behavior in Mice

A mouse model demonstrating strong and stable inheritable features of dominance and submissiveness was successfully established at Ariel University in Israel [185,270]. The populations of dominant (Dom) and submissive (Sub) mice were raised from the outbred Sabra strain, selectively bred for 48 generations based on the food competition Dominance Submissive Relationship (DSR) paradigm. The Sabra strain’s behavioral and biochemical characteristics lie within the diapason of those of C57BL/6, Balb/c and ICR mice [271]. The behavioral phenotypes of Dom and Sub mice were cross-validated in different behavioral tests including forced swim test (FST), resident-intruder test (RIT), three chamber test (TChT), and sucrose preference test and all show good face validity. These Dom and Sub mice react differentially to antidepressants, mood stabilizers and psychotropic agents and their inherited behavioral tendencies are dependent upon environmental and social triggers [180,181,182]. Paroxetine showed dose dependent anti-depressive-like effect in Sub mice and caused a paradoxical effect in Dom mice. Sub mice serve as an animal model of depression demonstrating depressive-like behavior and are susceptible to stressful stimuli, whereas Dom mice exhibit strong characteristics of manic-like behavior and show pronounced stress resilience [180,183]. The altered monoamine content in brain areas referring to emotionality and social hierarchy underly the behavioral phenotype in Sub and Dom mice. The decreased levels of 5-HT in the brainstem, reduced levels of norepinephrine in the prefrontal cortex and hippocampus, and elevated levels of dopamine in the prefrontal cortex, HPC, striatum and brainstem were measured in Sub mice [186].

Three-months adult Sub mice demonstrated impaired cognitive behavior, evaluated by 8-arm radial maze and by novel recognition test, with further deterioration in memory at 9 months [184]. This impairment in memory formation in Sub mice was correlated to abnormal overexpression of α-amino-3-hydroxy-5-methyl-4-isoxazolepropionic acid (AMPA) receptor and higher density of glutamate A1 (GluA1) in the hippocampus [184]. Transcriptomic analysis revealed differential hippocampal expression of genes involved in synaptic plasticity including Syn IIb isoform of SynapsinII gene, BDNF and IGF-1, which were significantly lower among Sub mice, in comparison to both Dom and wild type (WT) counterparts [184,185].

Proteomics analysis of hippocampal formation of Dom and Sub mice demonstrated expression changes in protein datasets responsible for social interaction. Among the most enriched categories, extensive changes were found in proteins related to presynaptic release, ion channel regulation, circadian rhythm, mitogen-activated protein kinase (MAPK), ErbB and NF-kB pathways [272].

Recent studies revealed that the innate stress susceptibility of Sub mice is reflected in significant reduction in the lifespan of both male and female. Shortened lifespan of Sub correlated with chronic inflammation, age-dependent splenomegaly and a significant increase in the circulating levels of IGF-1 and pro-inflammatory cytokines IL-1β and IL-6 [273].

The Dom and Sub mice fit well all the main demands of an animal model, demonstrating excellent etiological, face and predictive validity. The weakness of this model is that selective breeding resulted in strong augmentation and polarity of social phenotypic characteristics that may not necessarily be seen to such an extent in the human population.

#### 4.4.4. Selective Breeding Model of Low Activity in a Swim Test: High-Active (SwHi) and Low-Active (SwLo) Rats

Sprague-Dawley (SD) albino rats were selectively bred based on the swimming motor activity in the forced swim test. This breeding for at least 45 generations resulted in the development of two rat lines; Swim Low-Active (SwLo) and Swim High-Active (SwHi) rats [187]. SwLo rats demonstrated shorter struggling time and increased immobility phase, corresponding to depression-like behavior in rodents as feelings of despair. SwHi rats exhibited an opposite response: increased struggling time and shorter immobility time. Chronic, but not acute antidepressant treatment, reverses this depressive-like phenotype in SwLo rats [187]. Several chronically administered tricyclic antidepressants, such as imipramine, desipramine, venlafaxine, phenelzine and bupropion, efficiently prolonged the struggling time of SwLo rats in FST [189], but SSRIs were not effective. Both SwLo and SwHi rats exhibited similar locomotion activity tested by open field, and almost equal time spent in the open arm in the EPM study [188]. Therefore, SwLo rats rather represent the model of atypical depression [274]. It was suggested that depression-like behavior of SwLo rats in the swim test results from alteration in dopaminergic [190,191] and glutamatergic pathways [192]. This model has good face, etiological and predictive validity. Its weakness is that it relies mainly on differences in locomotor activity and therefore has a low construct validity.

#### 4.4.5. The Fawn-Hooded (FH/Wjd) Rat

Fawn-Hooded (FH/Wjd) rats are an inbred strain with abnormal serotonin storage in the platelets. Fawn-Hooded rats are associated with the hemorrhagic disorder, known as platelet storage pool deficiency, that resembles Chediak-Higashi syndrome in humans [193]. FH/Wjd rats exhibit a depressive-like phenotype with concomitant high voluntary ethanol intake and show good face validity as an animal model of depression, comorbid with alcoholism. These rats have low motor activity and increased immobility time in the forced swim test, accompanied by an elevated basal level of corticosterone. These features are reversed by chronic treatment with SSRIs [193,194]. Chronic fluoxetine treatment attenuates the immobility of FH/Wjd rats but does not affect alcohol intake [194]. Regarding construct validity, FH/Wjd rats exhibit abnormalities in function of the central serotonergic, GABAergic and HPA pathways [195,196,197,198].

### 4.5. Genetic Manipulation Induced Models of Depression

Mouse genetic models have played an important role in the elucidation of molecular pathways underlying human disease. The sequencing of the mouse genome revealed 90% gene homology to the human genome, with 80% identity among the protein-coding regions [275]. A widely used method to determine the function of a gene suspected of causation of a certain disorder, such as depression, is to delete the gene from the investigated organism (usually mice) by homologous recombination. Using null mutants or knockouts for that specific gene allows the investigation of the effect and the role of the absence of the specific gene (and the protein that it encodes) in the pathogenesis of certain diseases (such as depression). Another conventional strategy for gene modification, opposite to the development of knockout models, is transgenesis, when a foreign gene (i.e., human transgene or mutated gene) is introduced into a recipient organism’s genome, creating a transgenic animal.

Family-based studies in humans suggest about 40% of familiar heritability for depression [276]. Hence, genetic animal models may also help to better understand the role of heritability in depression. Genetic rodent models may provide proof of concept of the involvement of specific genes in depression pathophysiology.

Several molecular mechanisms were discovered to play a role in the pathogenesis of depression: among them are monoamine neurotransmitters such as serotonin [18,277,278,279], noradrenalin [18,280,281] and dopamine [282,283,284], and enzymes involved in their degradation (monoamine oxidase) or precursors of their synthesis (tryptophan) [279,281,282]. The most investigated genes are BDNF [17], serotonin receptor (5-HT) [285] and Sirt1 [286,287]. These genes regulate pathways that are implicated in the development of mood disorders; BDNF regulates neural plasticity and connectivity generally via tyrosine receptor kinase B (TrkB), and 5-HT, SLC6A4 and COMT are involved in the regulation of neurotransmitter signaling. The meta-analyses of genome-wide association studies (GWAS) suggested that FBXL4 (F-box and leucine rich repeat protein 4) and RSRC1 (Arginine and Serine Rich Coiled-Coil 1 or SRrp53) may play a role in MDD [288]. Additional genome-wide association data found that cAMP-specific 3′,5′-cyclic phosphodiesterase 4B (PDE4B) enzyme gene variants are associated with anxiety and stress-related disorders [289].

Based on the monoamine hypothesis of depression, the role of BDNF and genes involved in the HPA axis regulation, several models of knock-out/knock-in mice were generated (for detailed review see [199,290]). Here, we briefly describe some of the most commonly investigated genetically generated mice lines. The majority of these lines involve the inactivation of a candidate gene for depression, resulting in an anti-depressive-like phenotype. The deletions of the norepinephrine transporter, (NET-KO) mice demonstrated a stress resilience phenotype to restrained stress or social defeat [291]. The lack of 5-HT reuptake in SERT-KO mice causes the development of the phenotype that resembles the acute effects of selective serotonin reuptake inhibitor (SSRI) treatment, including increased anxiety, or even the serotonergic syndrome which is characterized by spontaneous dorsiflexion of the tail (Straub tail), tics, tremor and backward gait [292,293,294,295]. DAT-KO mice with dopamine transporter deletion developed depressive-like symptoms but have pronounced spontaneous hyperlocomotion [296,297] and mimic the effects of long-term exposure to psychostimulants. DAT-KO mice are, rather, a model of attention-deficit/hyperactivity disorder (ADHD), and their hyperlocomotion behavior was attenuated by treatment with Methylphenidate [298,299]. Increased expression of the 5-HT transporter in mice led to a low-anxiety phenotype [300]. Interestingly, mice with glutamate receptor gene GRIK4 deletion have reduced anxiety and an antidepressant-like phenotype [301]. Similarly, genetic deletion of the NMDA receptor NR2A subunit also results in decreased anxiety-like and depression-like behaviors [300]. Neurotrophic homozygous genetic models with deletion of BDNF and TRKB receptor are not viable [302,303,304], whereas heterozygous BDNF and TRKB receptor KO mice did not demonstrate differences from normal animals [305,306,307].

Genetic manipulation based on alterations of the HPA axis also provide a great opportunity to explore the molecular basis of the HPA axis in depression. Corticotropin-releasing factor (CRF) overexpression in mice results in increased anxiety-like behavior [308], but not depressive-like behavior [309]. Homozygous CRF knockout mice, despite being viable, do not demonstrate behavioral abnormalities compared to control littermates [310].

The use of Cre-driver mouse lines for targeting local specific brain areas associated with depression and emotionality such as forebrain, hippocampus and VTA, provides a new approach to investigate proof of concept as to how the brain modulates mood and mood disorders [311,312]. Within the development of Clustered Regularly Interspaced Short Palindromic Repeats (CRISPR)-associated protein 9 (Cas9) system (CRISPR/Cas9) technology, the new generation of KO mice were developed in order to investigate the role of specific proteins on depression-like phenotype development and for further translation to clinical depression [313,314,315].

Using genetically engineered mice provides an excellent platform to prove the concept of gain in function or loss of function of a specific gene. However, the generated line demonstrated an uncertain face and predictive validity [199,290]. Depression is a multifactorial disease underlined by multiple genes, and, therefore, single gene deletion or overexpression cannot overlap all depression core symptoms [199,290]. Conventional genetic deletion of genes results in spreading lack and loss of function throughout the body systems and induces impairments in many metabolic and system functions that are not necessarily seen to such an extent in human depression. The generation of knock-in or knock-out mice is also labor and cost intensive.

### 4.6. Is There an Ideal Animal Model of Depression?

The above described rodent models of depression indeed mimic many aspects of human MDD but none of them is “the ideal model”. Models in primates are generally closer to the human situation as the animals show similar symptomatology and often share a similar etiology [316,317,318]. Indeed, several models were developed in monkeys that might better fit the concept of an ideal model of a human disease. For example, Hennessy et al. showed that social isolation in male rhesus monkeys induced within several weeks a depressive-like posture and behavior in up to 90% of these monkeys [317]. A naturally occurring depression similar to that in humans was also described by Xu et al. in colonies of cynhomologous monkeys [318]. Hence, there seems to be an advantage in the use of monkeys as models that mimic human MDD. However, since it is not easy to establish primate colonies for such studies, the concomitant use of several rodent models, each one with its strengths and weakness, may overcome the lack of a single ideal model.

## 5. Conclusions

We described the more common models of depressive-like behavior in rodents. The fact that different facets of depressive-like behavior can be induced by genetic, environmental, surgical or pharmacologic means emphasizes the complexity of depression. It is generally observed that these models affect different brain neurotransmitters, further demonstrating that depression is not just an imbalance between monoamine neurotransmitters. While there is no single model that mimics all aspects of the human disease, the concomitant use of several of these animal models may be sufficient to cover all the different aspects of depression. As for future directions, there is still a need to develop a non-primate model that will resemble human depression in all aspects: etiology, pathogenesis treatment and prevention. Such a model must take into consideration the multiple etiologies of depression, the variety of clinical presentations and the fact that, often, current treatment modalities are ineffective. We are in particular need of models where we will be able to test exploratory treatment methods to enrich effective therapy. We should not forget, however, that the ultimate testing is in humans.

## Figures and Tables

**Table 1 diagnostics-11-00123-t001:** Strengths and weakness of animal models of depression.

Depression Models	Strengths	Weaknesses
Environmentally induced Models of depression		
1. Early-life stress		
Prenatal stress	Induce depression and anxiety-like phenotype: prolonged immobility (despair-like behavior) in FST and TST tests [58,59,60], anhedonia- reduced preference to sucrose solution [58]. and anxiety in EPM [58,61,62]. Induce changes along the hypothalamic-pituitary-adrenal (HPA) axis [60,63]. Useful in the research and elucidation of the epigenetic mechanisms underlying the consequences of gestational stress or early-life stress to later depression and anxiety.	Differences in the rate and time of brain ontogeny between rodents and humans [64,65,66]. Predictive validity of antidepressant treatment is uncertain. Difference in the rate and time of postnatal brain development between rodents and human
Maternal separation	Induce learning and memory deficits [67]; depressive- and anxious like behavior in open field and EPM [67,68]. Induce changes along the hypothalamic–pituitary–adrenal (HPA) axis [69,70,71]. Induces long-lasting behavioral changes until adulthood. Suitable for studying the interaction between genes and environment in a newborn animal.	Long-lasting behavioral changes are inconsistent due to different methodologies among research groups [72], depending on the mouse strain [73]; Using infant pups whose brain developmental status corresponds to prenatal human stage [64,65,66]. Poor predictive validity for antidepressant treatment at adulthood [74].
Post-weaning social isolation stress	Induce depressive- and anxious-like behavior in open field and EPM [75,76,77,78,79,80,81]. Demonstrated good predictive validity to antidepressants treatment [82]. Useful to study the effects of social isolation stressors on anxiety, depression and substance use disorders (SUDs) in adolescence [83,84,85,86]	It is rather a model of reward-associated behaviors such as drug preference and alcohol consumption in adulthood and other varieties of psychiatric disorder, such as schizophrenia and bipolar disorder [76,81,85,87,88].
2. Adulthood stress		
Social isolation	Induces both anxiety- and depression-like behaviors in FST, TST EPM, and sucrose preference test [89,90,91,92,93,94]. Demonstrated good predictive validity to antidepressants treatment [93]. Induce changes in HPA [92,93].	Study reports are inconsistent; isolation has strong strain-dependent and test-specific effects [95,96,97].
Learned helplessness rats/mice. (Acute stress)	Induces anxiety- and depression-like behaviors with good similarity to the symptoms of depression with cognitive and neuroendocrine impairments [98,99]. Induces changes in HPA [100]. Useful to investigate the effect of unpredictable and uncontrollable stress. Demonstrates good predictive validity for antidepressant treatment [101,102,103].	Inconsistent response [100]. Short-term effect. Weak etiologic validity because the exact role of the genetic construct of the stress-resilient and stress susceptible animal is as yet unknown
Chronic mild stress (CMS)	Induces both anxiety- and depression-like behaviors in FST, EPM, and sucrose preference tests [104,105,106,107,108,109]. Demonstrates good predictive validity for antidepressants treatment [110]. Induces changes in HPA [111]. Useful to investigate the effect of mild unpredictable stress.	Varied reproducibility of this model among different research centers [105].
Repeated restraint stress (CRS)	Induces depressive-like phenotype in social interaction, anhedonia, increased anxiety-like behavior, and impaired spatial learning [112,113,114,115]. Demonstrates good predictive validity for antidepressants treatment [112,116]. Induces changes in HPA [115]. Useful to explore the effect of chronic psycho-emotional stress [116,117,118].	Induced short-term depression symptoms tend to subside [112,113,114,115].
Chronic social defeat stress	Induces long-term depressive-like phenotype expressed as anhedonia and social avoidance [119,120,121,122]. Demonstrates good predictive validity for antidepressant treatment [123,124]. Induces changes in HPA and dopaminergic Ventral Tegmental Area (VTA) [122,125]. Useful to investigate the effect of repeated, high-pressure social stress on depression.	Requires the use of a large number of animals, takes up a lot of space in the animal facility and is also quite labor intensive. Requires to prevent significant damage to the intruder mice to avoid ethical concerns. Cannot model female animals.
Social instability stress (SIS)	Induce both anxiety- and depression-like behaviors in OF, EPM, sucrose preference and social interaction tests [126,127,128,129]. Demonstrates good predictive validity for antidepressant treatment [126]. Induces changes in HPA [126]. Useful to investigate the effect of repeated social stress on depression.	Inconsistent reports due to wide methodological variability among research groups [126].
Lesions of the brain that may result in depression		
Olfactory bulbectomy (OBX)	Induces both anxiety- and depression-like behaviors in OF, EPM, sucrose preference, social interaction tests and memory dysfunction [130,131,132,133,134,135,136,137,138,139,140]. Demonstrates good predictive validity for antidepressant treatment [133,135,141,142]. Induces changes in altered function of the noradrenergic, serotonergic, cholinergic, γ-aminobutyric acid, (GABA)ergic and glutamatergic neurotransmitter systems [130,143,144]. Useful in investigation of the chronic psychomotor agitated depression.	Injury to the olfactory bulbs cortex. Mimics only a limited part symptoms of depressed patients.
Pharmacological models		
Reserpine induced depression model	Induces depression-like behaviors in OF, FST, TST and sucrose preference tests [145,146,147,148]. Demonstrates good predictive validity for antidepressant treatment [147,149,150,151,152,153,154,155].	Produces motor impairment mimicking Parkinson disease dyskinesia, increased nociceptive sensitization, hypothermia and has high rate of mortality [148,155,156,157].
Corticosterone Model of Depression	Induces both anxiety- and depression-like behaviors in OF, EPM, sucrose preference, cognitive impairments in Morris water maze and T-maze tests [56,158,159,160]. Induces wide-range changes in stress-sensitive brain regions [40,160]. Demonstrates good predictive validity for antidepressant treatment [161,162,163].	May induce many metabolic and biochemical changes outside the brain, affecting animal behavior in a different way compared to human depression.
Genetic models		
Wistar Kyoto (WKY) model	Demonstrates depression-like behaviors in FST, OF, sucrose preference test [164,165,166,167,168]. Has innate HPA abnormalities [169,170,171,172].	Poor predictive validity for antidepressant treatment [173,174,175]. Exhibits significant differences in comparison to the human situation as it presents an abnormal HPA function [169,170,171].
Genetically-selected Flinders Sensitive Line (FSL) rat model	Demonstrates depression-like behaviors in FST [176]. Demonstrates good predictive validity for antidepressants treatment [177,178]. Demonstrates changes in the cholinergic, serotonergic, dopaminergic, neuropeptide Y, and circadian rhythm systems and normal HPA [177]. Useful in studies of human depression with psychomotor retardation and to evaluate efficacy and safety of antidepressant agents in pregnant dams and their offspring during early life and later at adulthood [179]	There are many changes in different brain neurotransmitters which are not necessarily present in people with depression.
Model of selective breeding for depressive-like (submissive) and manic-like (dominant) behavior in mice	Demonstrates both anxiety- and depression-like behaviors in OF, EPM, sucrose preference and social interactions tests, cognitive impairments in 8-arm maze and object recognition tests [180,181,182,183,184]. Demonstrates good predictive validity for antidepressants treatment [180,181,182]. Demonstrates changes in stress-sensitive brain areas [184,185,186]. Useful to study social hierarchy effect on depression and in investigations of stress resilience and stress susceptibility.	Selective breeding by Dominance–Submissive Relationship (DSR) test resulted in strong augmentation and polarity of social phenotypic characteristics that may not necessarily be seen to such extent in the human population.
Selective breeding model of low activity in a swim test: high-active (SwHi) and low-active (SwLo) rats	Demonstrates both anxiety- and depression-like behaviors in FST, OF and EPM tests [187,188]. Demonstrates good predictive validity for some antidepressants treatment [189]. Demonstrates changes in dopaminergic and glutamatergic pathways [190,191,192].	It relays mainly on differences in locomotor activity and, therefore, has a low construct validity.
The Fawn-Hooded (FH/Wjd) rat	Demonstrates depressive-like phenotype with concomitant high voluntary ethanol intake [193,194]. Demonstrates good predictive validity for some antidepressant treatment [194]. Demonstrates changes serotonergic, GABAergic and HPA pathways [195,196,197,198].	Association with the hemorrhagic disorder, known as platelet storage pool deficiency, that resemble Chediak-Higashi syndrome in humans [193] and therefore does not represent “pure” depression.
Genetically manipulated models of depression	Demonstrates good platform for proof of concept of gain in function or loss of function of particular genes.	Uncertain face and predictive validity [199]. Conventional genetic deletion of genes results in lack and loss of functions throughout the body systems and induces impairments in many metabolic and system functions that are not necessarily seen to a similar extent in human depression. Labor and cost intensive.

**Table 2 diagnostics-11-00123-t002:** Timing differences in brain development in mice and rats compared to human.

	Mice/Rats	Human
Neural tube formation and closure [65,66]	GD * 9–11	GD 20–28
Neurogenesis in the spinal cord and hindbrain structures [65,66]	GD 10	week 4 of gestation
Organogenesis [65,66].	GD 12–16	Week 8–27
Neurogenesis in the cerebellum and hippocampus. cell migration, myelination, and synaptogenesis [65,66].	First postnatal week-10 days	Late second and third trimester
Adolescence	PND ** 21–70	11–21 years

* Gestational Day. ** Postnatal Day.

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
