# Peer review of "Animal Models of Depression: What Can They Teach Us about the Human Disease?"

_diagnostics, 2021, doi:10.3390/diagnostics11010123_

Round 1

Reviewer 1 Report

This review is a summary of the state of the art in animal models  of depression-like behaviour. overall, the authors improved the review by taking out some of the negative vibe they had in the original manuscript. what i didnt understand is why they have also taken out some of the models they originally described.

But as i previously stated, there has been a wealth of recent reviews on animal models in depression and i am sorry to say i dont see the added benefit of having yet another one. what i am still missing is a new angle in their review, a suggestion for a new improved model, something that goes beyond the pure description of what is available.

Author Response

We thank the reviewer for the constructive comments. We hope that our respective changes/additions that indeed improve the manuscript are now satisfactory.

  1. “Why the authors have taken out some of the models they originally described.” We are indeed sorry for this mistake and we now added all models we previously described and added one additional new model of social isolation stress.
  2. ”A suggestion for a new improved model.” We added a new paragraph headed “Is there an ideal animal model of depression?” where we discussed the facts that primate models might be more appropriate than rodents. We now slightly modified this paragraph to add that the weakness of rodent models can be overcome by the concomitant use of several rodent models. This is also stated in the conclusions.

Reviewer 2 Report

In this manuscript entitled “Animal models of depression: what can they teach us about the human disease?”, the authors reviewed the most common animal models of depression and the behavioral tests used to analyze the depressive-like phenotype in rodents. The manuscript is well organized, although there are several track change corrections, and it is not always easy to follow the content. I recommend the publication of this manuscript after some revisions. Below are my suggestions.

In the abstract section (lines 22-26), the sentence “Most animal models are very relevant since they mimic many of the features observed in the human situation and may serve as a powerful tool for the study of the etiology, pathogenesis, and respond favorably to the chronic antidepressant treatment of depression. This is especially relevant since only few respond to acute treatment” should be included. 

In section “Testing of depression-like behavior in animals” the author should include also the splash test, a test widely used to evaluate motivational and self-care behaviour. 

The 4.1 paragraph “Prenatal stress modeling should” be included in the manuscript. Moreover, it would be useful to add a section regarding the Social isolation stress, either during adolescence and in adulthood, as another widely used model to induce depressive-like phenotype in animals. 

Increasing evidence suggests that stressed animals can be divided into susceptible and resilient, in particular using the CMS and the social defeat paradigms. This point should be more elaborated because replicates a fundamental aspect present in the human: not all the person responds similarly to stress exposure. This could be useful to identify the mechanisms underlying the stress susceptibility.   

Author Response

We thank the reviewer for the constructive comments. We hope that our respective changes/additions that indeed improve the manuscript are now satisfactory.

  1. In the abstract the sentence: “Most animal models are very relevant…” should be included”... We thank the reviewer for this. The sentence was included.
  2. “The authors should include also the splash test”. We added this test in section 3.2.
  3. “Prenatal stress modeling should…be included…” We added in section 4.1 the social isolation stress method as suggested by this reviewer.
  4. “Stress animals can be divided into susceptible and resilient…” We elaborated on this issue in section 4.1.1.4 and especially in 4.1.” Environmentally induced models of depression”

Reviewer 3 Report

In this manuscript, the authors first briefly introduced current knowledge about human depression, and reviewed current animal models of depression, which includes environmental-induced, genetic, surgical, or pharmacologic models that recapitulate different disease phenotypes.  

The following concerns are raised:

  1. A diagram to overview the different models with their strengths and weakness may be preferred.  
  2. Line 92-93, authors may include references of animal models of depression via different approaches.
  3. Line 185-186, “The brain developmental status of rats and mice at the time of birth corresponds to that of the third trimester of the human fetus and is characterized by elevated synaptic plasticity”, why the time of birth in rodents corresponds to the third trimester of the human fetus, is this analogy based on the status of neurogenesis, or synaptic plasticity, or brain structural status? A diagram with references will be better to support this claim.  Also, a prenatal stress model may be included to explore the effect of early-life experience on the development of depression.
  4. For the genetic models, are there any models available with specific genes perturbed, or generated via currently wide-adopted CRISPR-cas9 genomic editing?
  5. For the conclusion section, future directions in the animal model and therapeutic development should be included.

Author Response

We thank the reviewer for their constructive comments. We hope that our respective changes/additions that indeed improve the manuscript are now satisfactory.

  1. “A diagram to overview the different models with their strength and weakness” We added a table (Table 1) outlining all models with their strength and weaknesses
  2. Lines 92-93: “references of animal models of depression…”
  3. Lines 185-186: The brain developmental status….” We added a table (Table 2) outlining the differences and similarities in brain development and also inserted additional text and references; (lines 185-201).
  4. “Are there any models available with specific genes….” We thank the reviewer for raising this issue. Initially we thought not to discuss genetic manipulations because they may also induce other metabolic and functional changes. However, following the reviewer’s comments, we added a section discussing models of gene manipulations –section 4.5, describing some models of transgenic mice. We also added this to our Table 1 discussing the strengths and weaknesses of these models.
  5. “Future directions for the conclusion section”. We added several sentences in the conclusions related to future directions in the development of animal models. This was also mentioned in the section on “ideal animal models”.

Round 2

Reviewer 1 Report

the authors addressed my concerns and suggestions sufficiently

however, there seems to be a problem with the references, e.g. 210 is in the text referred to as Hennesy et al, work in monkeys, and also Xu et al, in the reference list that is an article by Bobkova on neurotrophin p75 receptor, i guess this is a glitch in the reference program file? easy to resolve

Reviewer 2 Report

The authors have answered all of my concerns.

This manuscript is a resubmission of an earlier submission. The following is a list of the peer review reports and author responses from that submission.

Round 1

Reviewer 1 Report

08.10.2020

In this ms, the authors have aimed to provide an overview of some of currently available models of pre- clinical models and tests for Major depressive Disorder.  As they have formulated the goals of the paper “ The purpose of the present review is to BRIEFLY discuss the more common animal models of depression and stress their strength and weakness“. Obviously, modeling depressive disorder in animals is fairly broad and complex area of research and it is problematic to imagine that such goal can be accomplished in a short paper. I would like to stress the following problems with this review:

  1. The paper does not use the terminology that is appropriate to be used while describing animal models of human diseases. The Authors do not operate such terms as “construct, face, predictive validity”, “pharmacological sensitivity” and “etiological relevance”. A comparison of discussed animals paradigms should be done in a frame of this commonly used terminology.  Instead, they operate conversational language in the most of the parts of the paper, using expressions like  “behavior traits  that can be considered analogous to the symptoms of depression in humans”, “resemblance”, "apathy-like behavior”, “understanding of scientific processes”,  etc, instead. Too many sections of the paper are written in rather non-academic conversational language.
  2. It is a mistake to state that the principles defining the validity of animal models were proposed by Willner; they were proposed years before him.
  3. Referencing needs to be improved by far, as many statements made, in particular in the first pages of the paper, which are not supported by appropriate citations. E.g., it is indicated that “Depression is probably the most common behavioral debilitating disease” but no specific information or references are given. This comment is applicable basically to the paper as a whole.
  4. In the beginning of the review the authors say that “… appropriate animal models can also help in the search  for effective treatment. Hence, both genetic epigenetic and environmentally induced animal models  that reproduce symptoms typical to human depression have been developed, especially in rodents, to help the investigators to better understand human depression“ However the conclusion made at the end of the paper as it is also expressed in the Abstract is in conflict with this statement and sounds rather negative „Animal models cannot serve for the study of prevention of depression”.
  5.  English has to be revised by a native speaker at least in terms of language style.

Reviewer 2 Report

This review is a summary of the state of the art in animal models  of depression-like behaviour. overall, i felt from reading the abstract, that the authors were extremely negative towards the validity of animal models to aid in investigating depression. i think anybody working in the field knows that there is not THE animal model for depression but rather animal models that can aid to mimick and investigate certain symptoms related with this devastating disease. saying that, there has been a wealth of recent reviews on animal models in depression and i am sorry to say i dont see the added benefit of having yet another one. not saying that the authors werent doing a good job listing tests that are undertaken, listing different models and their advantages and disadvantages of such that i am missing the new angle of their review, a suggestion for a new improved model, something that goes beyond the pure description of what is available.